# Encapsulation of Inositol Hexakisphosphate with Chitosan via Gelation to Facilitate Cellular Delivery and Programmed Cell Death in Human Breast Cancer Cells

**DOI:** 10.3390/bioengineering11090931

**Published:** 2024-09-17

**Authors:** Ilham H. Kadhim, Adeolu S. Oluremi, Bijay P. Chhetri, Anindya Ghosh, Nawab Ali

**Affiliations:** 1Department of Biology, Donaghey College of Science, Engineering, Technology, and Mathematics, University of Arkansas at Little Rock, 2801 South University Avenue, Little Rock, AR 72204, USA; ihkadhim@ualr.edu (I.H.K.); asoluremi@ualr.edu (A.S.O.); 2Department of Chemistry, Donaghey College of Science, Engineering, Technology, and Mathematics, University of Arkansas at Little Rock, 2801 South University Avenue, Little Rock, AR 72204, USA; bpchhetri@ualr.edu

**Keywords:** inositol polyphosphates, chitosan, encapsulation, apoptosis, cytotoxicity, reactive oxygen species, breast cancer

## Abstract

Inositol hexakisphosphate (InsP_6_) is the most abundant inositol polyphosphate both in plant and animal cells. Exogenous InsP_6_ is known to inhibit cell proliferation and induce apoptosis in cancerous cells. However, cellular entry of exogenous InsP_6_ is hindered due to the presence of highly negative charge on this molecule. Therefore, to enhance the cellular delivery of InsP_6_ in cancerous cells, InsP_6_ was encapsulated by chitosan (CS), a natural polysaccharide, via the ionic gelation method. Our hypothesis is that encapsulated InsP_6_ will enter the cell more efficiently to trigger its apoptotic effects. The incorporation of InsP_6_ into CS was optimized by varying the ratios of the two and confirmed by InsP_6_ analysis via polyacrylamide gel electrophoresis (PAGE) and atomic absorption spectrophotometry (AAS). The complex was further characterized by Scanning Electron Microscopy (SEM) and Fourier Transform Infrared Spectroscopy (FTIR) for physicochemical changes. The data indicated morphological changes and changes in the spectral properties of the complex upon encapsulation. The encapsulated InsP_6_ enters human breast cancer MCF-7 cells more efficiently than free InsP_6_ and triggers apoptosis via a mechanism involving the production of reactive oxygen species (ROS). This work has potential for developing cancer therapeutic applications utilizing natural compounds that are likely to overcome the severe toxic effects associated with synthetic chemotherapeutic drugs.

## 1. Introduction

Inositol polyphosphates (InsPs) are naturally occurring compounds present in both plant and animal cells. They play a major role in the regulation of various cellular processes, including calcium mobilization, vesicular trafficking, cell growth and differentiation, gene expression, and programmed cell death (PCD) [1,2,3,4,5]. Inositol 1,4,5-trisphosphate (InsP_3_) is the first water-soluble InsP produced in the cell from the membrane-bound inositol phospholipid, phosphatidylinositol 4,5-bisphosphate (PIP_2_), by phosphatidyl inositol-specific phospholipase C (PI-PLC) during G-protein-coupled receptor activation. It is a well-known signaling molecule for calcium mobilization and serves as a precursor for all other cytosolic InsPs. There exists a complex InsP metabolic pathway [6,7] regulated by the phosphorylation and de-phosphorylation of InsP_3_, giving rise to more than 60 InsPs metabolites. Among various InsPs, inositol hexakisphosphate (InsP_6_) is the most abundant InsP and is present in much higher amounts in plant than animal cells. In plants, it is also referred to as phytic acid (IP_6_), which is primarily enriched in major food grains, nuts, and legumes, where it serves as a phosphate reservoir. It also serves as an antioxidant to prevent oxidative damage and is thus regarded as a key nutrient in human diet [8,9,10,11].

Mammalian cells have their own potential to synthesize InsP_6_ from cytosolic InsP_3_ [1,12], albeit in smaller amounts. Previous studies, including work from our laboratory, have demonstrated that exogenously administered and endogenously manipulated InsP_6_ induces apoptosis in breast and other cancer cells [3,13,14,15]. InsP_6_ binds to the pleckstrin homology (PH) domain of AKT of the PI3K/AKT/mTOR signaling pathway [15,16] to inhibit cell proliferation and induce apoptosis. The induction of apoptosis is specifically associated with the presence of phosphate groups on the inositol ring, and replacing phosphate with sulfurate to produce inositol hexakisphosphate (InsS_6_) inactivates its apoptosis-related activity [3]. It has also been reported to inhibit cell growth and induce G1 phase arrest in prostate cancer cells [17]. In colon cancer cells, InsP_6_ inhibits cell proliferation and induces apoptosis by suppressing the PI3K/AKT/mTOR signaling pathway [18]. The use of InsP_6_ as a dietary supplement to prevent and treat cancer is well documented in the literature [19]. Furthermore, this naturally occurring compound has virtually no known toxicity, which is a desirable feature for any cancer-preventative agents. This feature has prompted many commercial vendors to sell InsP_6_ and its derivatives as nutritional supplements for public consumption to prevent and treat cancer [19]. These peculiar characteristics possessed by InsP_6_ make it a highly advantageous anticancer agent [19].

Cancer cells exhibit distinct attributes across tumor types which enable them to grow and metastasize to other organs [20,21,22]. Despite many advances made in cancer therapy, the mortality rate remains high, especially for breast cancer, which is a leading cause of deaths among women worldwide [23,24]. Cancer cells evade PCD or apoptotic mechanisms by accumulating high levels of anti-apoptotic molecules and low levels of pro-apoptotic molecules [25,26]. A successful therapeutic strategy requires inactivating cancer cells by inducing apoptosis without damaging normal cells to avoid necrosis, or cell death by injury [27,28]. Apoptosis is a key developmental process to maintain tissue homeostasis that safely disposes of dead cells by phagocytosis without triggering inflammatory responses [29,30]. So, it is crucial for a cancer therapeutic drug to induce the apoptotic cell death process. Apoptotic cell death is characterized by nuclear condensation, loss of mitochondrial membrane integrity, generation of reactive oxygen species (ROS), activation of caspase cascade, and the release of apoptotic biomarkers such as cytochrome C and ATP [31,32,33,34,35].

It is well established that exogenously administered InsP_6_ induces apoptosis in cancerous cells and therefore has great potential in the prevention and treatment of cancers [36]. However, because of the highly negative charge on InsP_6_, its entry across semipermeable plasma membrane is hindered. This has prompted us to develop a strategy to deliver InsP_6_ into cancer cells by encapsulating it with chitosan (CS), a naturally occurring, biodegradable, positively charged polysaccharide, to shield off the negative charge on InsP_6_ and study its effect on apoptosis. Our hypothesis is that the encapsulated InsP_6_ will enter the cell more efficiently and induce more apoptosis as compared with equivalent concentrations of free InsP_6_ (non-encapsulated). To the best of our knowledge, this work represents a novel InsP_6_ delivery mechanism to trigger apoptosis in breast cancer cells for the first time. However, attempts were made to deliver negatively charged inorganic polyphosphates (PolyPs) using amphipathic guanidinium-rich polycarbonates as nanocarriers, with an emphasis on understanding their biological relevance [37]. Other negatively charged molecules such as microRNA [37] and SiRNA [38] were delivered using lipid-based nanocarriers. More recently, the uptake and efflux of inorganic phosphorous (Pi) were studied in relation to cellular Pi homeostasis [39,40,41] where the uptake and efflux of Pi were found to be mediated by a xenotropic and polytropic retrovirus receptor 1 (XPR1), a ubiquitous transporter known to maintain cellular Pi homeostasis [39]. Interestingly, in this context, cellular levels of Pi were found to be modulated by cellular levels of higher InsPs, inositol pyrophosphates (PP-InsPs), particularly InsP_8_ [41]; thus, InsP_8_ was regarded as a Pi sensor in this study.

Herein, CS is used to encapsulate negatively charged InsP_6_ to facilitate its cellular delivery. Recently, CS has attracted a great deal of attention in pharmaceutical and biomedical fields. Originally, CS was called a marine polysaccharide as it was generated by the partial deacetylation of chitin from shrimps and crabs [42,43,44]. Nowadays, it is used as one of the most suitable biological materials for biomedical application because of its excellent biocompatibility, biodegradability, non-toxicity, and adsorption characteristics [44]. Several studies have indicated that CS is a good carrier to encapsulate negatively charged molecules for drug delivery purposes [43]. For example, Lee and his colleagues [44] encapsulated insulin with CS to achieve stable oral delivery of the insulin. Sachdeva and his colleagues similarly used chitin and chitosan biopolymers for drug delivery in cancer treatment [45]. In addition, the encapsulation of 5-fluorouracil [46] and oral formulation [47] with CS has been shown to be a smart drug delivery agent for possible application in cancer treatment. CS has also been grafted to a series of hydrophilic amino acids to develop a biocompatible amphiphilic drug delivery vehicle [48]. Additional studies have indicated that coating CS on iron oxide resulted in decreased cellular damage [49,50]. The significance of CS in this work is to use the positively charged biodegradable and membrane permeable polymer to encapsulate the highly negatively charged InsP_6_ to form the CS:InsP_6_ nanomaterial complex. This complex facilitates its entry through the cell membrane and releases InsP_6_ there to induce apoptosis (Figure 1).

## 2. Materials and Methods

### 2.1. Materials

Deacetylated (85%) chitosan (CS) with a molecular weight range of 100–300 kDa was obtained from Alfa Aesar Chemicals, Haverhill, MA, USA. The human breast cancer MCF-7 cell line, Dulbecco’s Modified Eagle Medium (DMEM), and Fetal Bovine Serum (FBS) were obtained from the American Type Culture Collection (ATCC, Manassas, VA, USA). D-myo-inositol hexakisphosphate (InsP_6_), Phosphate-Buffered Saline (PBS, pH 7.4), 2′,7′-dichlorodihydrofluorescein diacetate (DCFH-DA), acridine orange, ethidium bromide, dimethyl sulfoxide (DMSO), Tween-80, and etoposide were from Sigma-Aldrich, St. Louis, MO, USA. 3-(4-,5-dimethylthiazol-2-yl)-2,5-diphenyltetrazolium bromide (MTT), 10,000 U/mL penicillin, 10,000 U/mL streptomycin, and trypsin–EDTA (0.25%) were purchased from Fisher Scientific, Hanover Park, IL, USA. The Vybrant Flow Cytometry apoptosis assay kit was obtained from Invitrogen, Carlsbad, CA, USA. All other chemicals and reagents were of the highest analytical grade available and used before their expiry dates. Double-distilled deionized water was used to prepare all solutions.

### 2.2. Encapsulation of InsP_6_ with Chitosan

The CS:InsP_6_ complex was synthesized based on the ionic gelation method as described previously, with slight modifications [47]. Various studies have described the preparation of chitosan nanomaterial using tripolyphosphate (TTP) as a cross-linking agent. TPP is a non-toxic polyanion that interacts with chitosan via electrostatic forces to form an ionically cross-linked network. InsP_6_ is a naturally occurring substance used as a novel cross-linking agent in this study. The number of anions in InsP_6_ that can react with cations on chitosan becomes two-fold higher under acidic conditions than in TPP [47]. Hence, acidic conditions were employed to help dissolve chitosan using acetic acid. Briefly, CS was dissolved in 1% (*v*/*v*) acetic acid to produce chitosan at concentration of 5.0 mg/mL. InsP_6_ was dissolved in distilled water at 5.0 mg/mL. All solutions were filtered through a 0.22 μm membrane filter (Millipore Sigma, Burlington, MA, USA). The CS solution was flush-mixed with InsP_6_ solution at volumetric ratios of CS:InsP_6_ of 0.5:1.0, 1.0:1.0, 2.0:1.0, and 2.5:1.0 (*v*/*v*) under magnetic stirring at room temperature followed by addition of 0.5% (*v*/*v*) Tween-80 as a dispersive agent to prevent nanoparticle aggregation during gelation process. The pH of the solution was adjusted to a 4.6–4.8 range by adding 1N NaOH. The formation of the CS:InsP_6_ nanomaterial complex started spontaneously. After the completion of gelation, it was separated out by centrifugation at 12,000× *g* for 30 min, followed by three washings with ethanol to remove any acetic acid residues. Finally, the complex was freeze-dried to obtain the nanomaterial complex in dry form and used for further analysis after resuspension in deionized water. Figure 2 shows the schematic process for the preparation of the CS:InsP_6_ nanomaterial complex by the ionic gelation method.

### 2.3. Scanning Electron Microscopy (SEM)

The surface morphology of the dried CS and CS:InsP_6_ samples was examined by Scanning Electron Microscopy (SEM) using a JEOL (JSM 7000F Joel, Peabody, MA, USA) scanning electron microscope. For sample preparation, completely dehydrated samples of CS and CS:InsP_6_ were placed on a small piece of carbon tape supported on SEM specimen aluminum stubs to make the samples electronically conductive. The SEM images were captured at different magnifications to investigate any changes in the surface morphology of the CS following encapsulation of InsP_6_. The morphology of the particles was imaged in a secondary and backscatter electron mode under vacuum with accelerating voltage of 15 kV [47,51].

### 2.4. Fourier Transform Infrared (FTIR) Spectra Studies

Further characterization of the CS:InsP_6_ nanomaterial complex was performed by Fourier Transform Infrared Spectroscopy (FTIR) using Nicolet 6700 (Thermo Scientific, Waltham, MA, USA) FTIR spectrometer equipped with a DLaTGS detector and a XT-KBr beam splitter. The FTIR samples were prepared by using the standard KBr pellet method and the spectrum was recorded in the range of 4000–400 cm^−1^. The background was run with CO_2_ and water [51].

### 2.5. Detection of InsP_6_ by Polyacrylamide Gel Electrophoresis (PAGE)

InsP_6_ contents in the encapsulated CS:InsP_6_ complex or in cell extracts following cellular uptake studies were determined by polyacrylamide gel electrophoresis (PAGE), which separates various inositol polyphosphates based on their charge and size [52]. For the determination of the cellular uptake of InsP_6_, titanium dioxide (TiO_2_) was used to extract InsP_6_ from the cells [53]. Briefly, MCF-7 cells were cultured in a T-75 flask using DMEM and the cells were treated with free InsP_6_ (1.0–4.0 µM) or equivalent concentrations of the CS:InsP_6_ complex. The treated cells were incubated with TiO_2_ (5 µM), harvested by trypsinization, and washed twice with 1 × PBS (pH 7.4) by centrifugation at 14,000× *g* for 5 min. The resulting pellet was mixed with 100 µL of perchloric acid (1M) and kept on ice with continuous vertexing for 10 min and then incubated with 50 µL of 10% ammonium hydroxide on ice for 30 min. It was then centrifuged at 14,000× *g* for 5 min. The supernatant containing InsPs was transferred to a new Eppendorf tube and the pH of the solution was adjusted to 6.8. Various concentrations of standard InsP_6_ were run in parallel.

In this study, the samples were run on a 1.5 mm thick mini gel comprising 33% polyacrylamide in Tris-Borate-EDTA (TBE) (7.925 mL 40% acrylamide/bis-acrylamide (19:1), 0.950 mL 10× TBE, 0.550 mL dH_2_O, 67.5 μL 10% APS and 7.5 µL TEMED, Tetramethylethylenediamine). The samples were mixed with 6X Blue/Orange Loading Dye [0.4% orange G, 0.03% bromophenol blue, 0.03% xylene cyanol FF, 15% Ficoll 400, 10 mM Tris-HCl (pH 7.5) and 50 mM EDTA (pH 8.0)] prior to loading onto the gel. The gel was run at 100 volts for about 2 h at room temperature or until the orange dye reaches close to the bottom edge of the gel. Following PAGE, the gels were gently agitated for 30 min at room temperature in a filtered staining solution (20% methanol; 2% glycerol; 0.05% toluidine blue). Then, the gel was destained with 4–5 changes of destaining solution to remove any unbound toluidine blue dye. The photographs were taken after exposing the gel to white light on a transilluminator. The band intensities were analyzed using ImageJ software version 1.54 and expressed as arbitrary units. InsP_6_ amounts in unknown samples were calculated based on the band densities of the known concentrations of standard InsP_6_.

### 2.6. Detection of InsP_6_ by Atomic Absorption Spectrophotometry

The principle of this analysis is based on the measurement of elemental phosphorus composition in given amounts of chitosan-encapsulated samples by atomic absorption spectrophotometry (AAS) analysis and then converting phosphorous contents into InsP_6_ contents based on their molar equivalency. The analysis was performed commercially by Midwest Microlab, Indianapolis, IN, USA. Briefly, known amounts of solid samples were subjected to complete digestion in a micro bomb chamber using appropriate reagents. The apparatus was clamped shut and heated in a flame for 60 s. The mix was then cooled, washed and filtered. The contents were then diluted appropriately for further analysis. This procedure derivatized the processed sample in the form of acetone phosphate complexes with unique golden-yellow color. The data were supplied in two formats: as phosphorous percentage and milligrams of phosphorous per given amount of the CS:InsP_6_ complex. The amounts of InsP_6_ in the CS:InsP_6_ complex were then calculated based on the molecular weights of Pi and InsP_6_ and compared with the amounts of InsP_6_ determined by PAGE.

### 2.7. Cell Culture

Human breast cancer MCF-7 cells were grown in a complete Dulbecco’s Modified Eagle Medium (DMEM) containing 10% FBS, penicillin (500 Units/mL), and streptomycin (500 Units/mL) and maintained in a humidified CO_2_ incubator set at 37 °C with 5% CO_2_ atmosphere, as per manufacturer recommendation and modified as described earlier [54,55,56]. For subculturing, cells grown to 80% confluence were washed with PBS (pH 7.4) and then detached with trypsin-EDTA buffer pH 7.4 (0.05% trypsin/0.53 mM EDTA in Hank’s Balanced Salt Solution (HBSS) without sodium bicarbonate, calcium, and magnesium). Subsequently, trypsin was removed by centrifugation and the cells resuspended in fresh DMEM complete medium. Cell density was determined by counting the viable cells using a hemocytometer following staining with 0.4% Trypan blue dye [54].

### 2.8. MTT Assay

The cytotoxic effects of the CS:InsP_6_ complex were assessed by MTT assay [54]. In brief, MCF-7 cells were seeded in 96-well plates (Costar, Sigma-Aldrich, Inc., St. Louis, MO, USA) at a density of 1.0–1.5 × 10^4^ cells per well in 100 µL cell culture growth medium and grown for 24 h at 37 °C with 5% CO_2_. Cells were then treated with various concentrations of the CS:InsP_6_ complex and free InsP_6_ and CS as vehicle control and incubated further for indicated time intervals. At the end of the incubation period with the InsP_6_ complex and CS, the medium was aspirated, and the cells were washed twice with PBS and suspended in 90 μL cell culture medium containing no serum. A volume of 10 μL MTT solution (5 mg/mL) was then added to all wells and cells were incubated for an additional 4 h in the dark. Thereafter, the medium was aspirated, and the cells were lysed in 100 μL of dimethylsulfoxide (DMSO) to dissolve the insoluble MTT formazan complex. The color thus developed was read at 570 nm using a microplate reader (Bio Tech SYNERGY H4, Santa Clara, CA, USA). The results were presented as percentage of the control values. All experiments were performed in triplicate, and each was repeated at least three times.

### 2.9. Acridine Orange/Ethidium Bromide Staining

Apoptosis was determined by fluorescence microscopy following acridine orange/ethidium bromide (AO/EB) staining using the established procedure reported previously [32,56]. Briefly, the cells were seeded in 6-well plates at a density of 15 × 10^3^ cells/well in 100 µL cell culture growth medium and grown overnight in complete medium at 37 °C with 5% CO_2_. The cells were then treated with different concentrations of CS, free InsP_6_, and the CS:InsP_6_ complex. Etoposide (100 µM) was used as a positive control. After treatment for given time periods, cells were washed with 1 × PBS (pH 7.4) and stained with a 1:1 ratio of the solution consisting of 100 mg/mL acridine orange and 100 mg/mL ethidium bromide in PBS. Following washing with PBS, cells were observed immediately under UV light by using an OLYMPUS BX51 fluorescence microscope equipped with a digital camera. Viewing fields were selected randomly and photographed. Acridine orange stains live cells green, whereas ethidium bromide stains fragmented nuclear DNA in dead cells red. The numbers of live (green) and apoptotic (red) cells were counted. Approximately 200 to 300 cells per treatment were counted to calculate the percentage of apoptotic cells using the following expression:(1)$$Percentage\;Apoptosis=\frac{Number\;of\;dead\;cells}{Total\;number\;of\;cells}\;\times 100$$

### 2.10. Determination of Reactive Oxygen Species (ROS) Production

Reactive oxygen species in MCF-7 cells were determined using 2′,7′-dichlorodihydrofluorescein diacetate (DCFH-DA), a redox-sensitive fluorescence probe, as reported previously [33,54]. The cells were seeded in black, clear-bottom, 96-well plates at a density of 1.5 × 10^4^ cells/well in 100 µL cell culture growth medium and grown overnight in complete DMEM at 37 °C with 5% CO_2_. The cells were then treated with the indicated concentration of the CS:InsP_6_ complex and equivalent amounts of free InsP_6_ for 24 h. At the end of the incubation, DCFH-DA was added to the medium at a final concentration of 10 µM and incubated for additional 30 min in the dark. Subsequently, the cells were washed twice with 1 × PBS (pH 7.4) to remove any unbound fluorescence probe and resuspended in 100 µL PBS. DCFH-DA staining intensity was determined using a fluorescence microplate reader (SYNERGY H4, Bio-Tek, Santa Clara, CA, USA, Hybrid technology) set at excitation and emission wavelengths of 485 nm and 535 nm, respectively.

### 2.11. Flow Cytometry

The flow cytometry assay was performed using Vybrant apoptosis assay kit #4 as described previously [55]. MCF-7 cells were seeded at a density of 6 × 10^5^ cells/mL cell culture growth medium in 6-well plates and grown overnight in complete DMEM at 37 °C with 5% CO_2_. Apoptosis was induced with indicated concentrations of free InsP_6_ and the CS:InsP_6_ complex, and etoposide was used as a positive control. The incubation continued for an additional time interval as indicated in Figure legends. The cells were then harvested by trypsinization and washed twice with 1x PBS (pH 7.4) and resuspended at a density of 1 × 10^6^ cells/mL in PBS. Next, 1 µL of YO-PRO-I stock solution (component A) and 1 µL of PI stock solution (component B) were mixed per milliliter of the cell suspension. YO-PRO-1 stock solution (component A) is a 100 µM YO-PRO-1 solution in DMSO. It is a green fluorescence dye permeant to apoptotic cell membranes. Propidium Iodide (PI) stock solution (component B) is 1 mg PI/mL (1.5 mM) solution in deionized water. PI is a red fluorescent dye that enters necrotic cells and gives red fluorescence. Both dyes are thus diluted 1000-fold. After 30 min of incubation at 4 °C, the cells were analyzed using a BD FasCalibular Flow Cytometer at the University of Arkansas for Medical Sciences Core Facility to sort out cell populations based on their differentially labeled fluorescence profile. Fluorescence emissions were measured at 515 to 545 nm for YO-PRO-I/FITC using an FL-1PMT detector and at 564 to 606 nm for PI using an FL-2 pre-menstrual Tension (PMT) detector. At least 10,000 cells were analyzed for each treatment condition and data were analyzed by Flow Jo software to detect different cell populations. This assay detects four types of cell populations shown in four quadrants. (1) Live cells (bottom left panel) show a low level of green fluorescence. (2) Early-apoptotic cells (bottom right panel) show an incrementally higher level of green fluorescence. (3) Late-apoptotic (top right panel) cells show both red and green fluorescence and (4) necrotic dead cells (top left panel) show increasing levels of red fluorescence.

### 2.12. Statistical Calculations

Statistical analysis was performed using GraphPad Prism v 7.0 software (San Diego, CA, USA). The results were expressed as means ± SD from three to four independent experiments each run in triplicate or as mentioned in figure legends. A one-way ANOVA test was used to determine values with significant differences between control and experimental groups. *p* values (*p* ≤ 0.05) were taken as values with significant differences.

## 3. Results

### 3.1. Encapsulation of InsP_6_ with Chitosan and Its Physicochemical Characterization

To facilitate the delivery of the negatively charged InsP_6_ inside MCF-7 cells, it was encapsulated with chitosan (CS) to shield off the negative charge by the ionic gelation method, as shown in Figure 2. The optimization of the incorporation of InsP_6_ in CS was studied by varying CS-to-InsP_6_ ratios during encapsulation. The extent of the encapsulation of InsP_6_ was determined quantitatively by PAGE analysis (Figure 3). The ratios of CS to InsP_6_ used were 0.5:1.0, 1.0:1.0, 2.0:1.0, and 2.5:1.0. Standard InsP_6_ with known concentrations were run in parallel to establish the linearity of InsP_6_ detection and to determine the InsP_6_ contents incorporated in the complex. The analysis of InsP_6_ band density showed (Figure 3A) a linear detection of standard InsP_6_ (R^2^ = 0.9998). Figure 3B shows the amounts of InsP_6_ incorporated in the CS:InsP_6_ complex were 0.27, 0.17, 0.14, and 0.49 μg for CS:InsP_6_ ratios of 0.5:1.0, 1.0:1.0, 2.0:1.0, and 2.5:1.0, respectively, in the samples loaded on the gel. The conversion of 0.49 µg InsP_6_ per given amounts of CS used indicated that a CS-to-InsP_6_ ratio of 2.5:1.0 gave the most incorporation of InsP_6_ (49 µg InsP_6_ per mg chitosan). This value was obtained by calculating the total InsP_6_ in the complex using the dilution factor of the sample loaded on the gel and per mg of CS. Although a definite trend of increased incorporation of InsP_6_ is not clearly seen with increasing amounts of CS, the values were more than the vehicle control (not shown in this representative experiment). The encapsulated complex with a ratio of CS:InsP_6_ of 2.5:1.0, which gave the maximum incorporation of InsP_6_, was used for further characterization of the complex by SEM, FTIR, and in cellular uptake studies.

We also used an alternative method to estimate InsP_6_ contents in the same encapsulated complex with a CS:InsP_6_ ratio of 2.5:1.0 by the determination of elemental phosphorous (Pi) by atomic absorption spectrophotometry (AAS) and converting Pi values to equimolar values in InsP_6_. AAS results showed that approximately 35 μg InsP_6_ was encapsulated per mg of chitosan, which was comparable to the value determined by PAGE (49 μg InsP_6_/mg chitosan). A lower estimation of InsP_6_ by AAS analysis might be due to differences in methodologies and the drastic acid and heat treatment procedures used in AAS analysis.

Following the confirmation of InsP_6_ incorporation into CS, physical characterization of changes in morphological appearance of the encapsulated complex was studied by Scanning Electron Microscopy (SEM). SEM analysis revealed that the encapsulated CS:InsP_6_ complex appeared filled and much denser and as whitish material compared to CS alone, which contained darker material with wide empty spaces (Figure 4). These results suggest morphological changes occurred in CS following ionic gelation, which allowed for the incorporation of InsP_6_ into the empty spaces seen in CS. Since SEM analysis does not exactly show any incorporation of InsP_6_ in chitosan, we next performed FTIR spectrum analysis, which confirmed the binding of InsP_6_ with chitosan.

To further study any changes in terms of chemical interaction between CS and InsP_6_ or the formation of new bonds as a result of encapsulation, FTIR spectrum analysis was performed on the encapsulated CS:InsP_6_ complex (ratio 2.5:1.0) and compared with that of CS and InsP_6_ alone (Figure 5). A characteristic O-H band at 3357 cm^−1^ in CS (Figure 5b) was shifted to 3399 cm^−1^ and became wider in the encapsulated CS:InsP_6_ complex (Figure 5c). These results indicated successful encapsulation of InsP_6_ in CS and an enhancement of hydrogen bonds following the complex formation. Furthermore, a characteristic band at 1140 cm^−1^ in the encapsulated complex indicated the presence of P=O group. However, this band was located at a slightly lower frequency (944 cm^−1^) in the FTIR spectra of InsP_6_ (Figure 5a). Additionally, the amide band of CS at 1648 cm^−1^ disappeared and shifted to 1629 cm^−1^ in the complex. The additional amide band stretching observed at 1598 cm^−1^ in CS was shifted to 1535 cm^−1^ in the complex indicating that an ionic cross-linking had occurred at the amine group of CS during encapsulation.

While the mechanism of the incorporation of InsP_6_ into CS may just be a simple interaction of negatively charged phosphate groups with positively charged CS moieties, the important aspect of this work is its efficient cellular entry and effect on apoptosis. The following studies are designed to establish cellular uptake of the encapsulated InsP6 and its action on inducing apoptosis.

### 3.2. Cellular Uptake of Encapsulated InsP_6_ by MCF-7 Cells

Following physicochemical characterization to confirm encapsulation of InsP_6_ with chitosan, the cellular entry of encapsulated InsP_6_ into MCF-7 cells was studied and compared with that of free (unencapsulated) InsP_6_. Prior to studying cellular uptake, we also determined a concentration dependent detection of InsP_6_ in the encapsulated CS:InsP_6_ complex. Using the encapsulated complex where InsP_6_ was incorporated the most (CS:InsP_6_ ratio: 2.5:1.0), an increasing amount of InsP_6_ was detected by PAGE proportional to the increasing volumes of the complex loaded on the gel (Figure 6A). This CS:InsP_6_ ratio was then used for all follow-up studies. Cellular uptake studies revealed (Figure 6B) that when equivalent concentrations of the encapsulated CS:InsP_6_ complex, free InsP_6_ and CS were incubated with MCF-7 cells, and a significantly increased amount of InsP_6_ was delivered into MCF-7 cells when it was encapsulated as compared to free InsP_6_, suggesting that CS is an excellent delivery vehicle for InsP_6_ into the cells. There was no InsP_6_ detected in the CS control.

### 3.3. Effect of Encapsulated InsP_6_ on Cell Viability in MCF-7 Cells

As evident from the MTT-based cytotoxicity assay, treatment of MCF-7 cells with encapsulated InsP_6_ increased cell death in a dose- and time-dependent manner (Figure 7). A significantly lower cell viability was observed at a 4 µM InsP_6_ equivalent of the encapsulated complex in 48 and 72 h compared with 4 µM free InsP_6_. Etoposide, known to induce apoptotic cell death, was used a positive control. Cell viability was drastically and significantly reduced under 100 µM etoposide within 24 h compared to untreated cells. There was no significant decrease in cell viability observed in the cells treated with equivalent concentrations of encapsulated InsP_6_ at lower concentrations (1.0 and 2.0 µM) as compared with respective concentrations of free InsP_6_ or CS at all the times tested. These results clearly indicate that encapsulated InsP_6_ can enter the cells and induce cell death in cancer cells.

### 3.4. Effect of Encapsulated InsP_6_ on Apoptosis in MCF-7 Cells

To determine whether encapsulated InsP_6_-induced cell death was due to apoptosis, a programmed cell death process, acridine orange/ethidium bromide staining was used to determine apoptotic and live cells using fluorescence microscopy. Representative fluorescence photomicrographs (Figure 8A) showed minimal sign of apoptosis in control (green cells), while maximum apoptosis (red cells) was seen in etoposide (positive control)-treated cells. Treatment with encapsulated InsP_6_ at 4 µM increased the amount of apoptotic (red) cells, while equivalent concentrations of free InsP_6_ or CS alone did not show any significant changes from green (control) to red (apoptotic) cells. Figure 8B showed a quantitative assessment of the percent apoptosis based on counting 200–300 cells from randomly selected photomicrographs of each treatment shown in Figure 8A. There was a significant increase in apoptosis in MCF-7 cells treated with the encapsulated InsP_6_ complex as compared to the control or treatment with the equivalent concentration (4 µM) of free InsP_6_ or CS (Figure 8B). Etoposide (100 µM) used as a positive control showed a robust increase in apoptosis.

### 3.5. Encapsulated InsP_6_ Induced Cell Death via the Generation Reactive Oxygen Species

Since the mitochondria are the primary originator of ROS production, we determined whether the CS: InsP_6_ complex affects ROS production. Mitochondrial generation of reactive oxygen species (ROS) is considered a key marker of the induction of the intrinsic apoptotic pathway. Herein, ROS were measured by 2′,7′-dichlorodihydrofluorescein diacetate (DCFH-DA). In consistent with the apoptotic cell death results shown in Figure 8, the ROS data showed that 4 µM equivalent of the encapsulated CS:InsP_6_ complex significantly increased the production of ROS as compared to free InsP_6_ or CS alone (Figure 9). Etoposide (100 µM), used as a positive control, also produced ROS significantly. These data clearly indicate that encapsulated InsP_6_ (CS:InsP_6_) is capable of inducing apoptosis via the production of ROS in MCF-7 cells.

### 3.6. Encapsulated InsP_6_ Induces Apoptosis

To confirm whether the encapsulated InsP_6_-induced cell death was mediated via inducing apoptotic cell death or merely due to nonspecific necrotic effects, flow cytometry analysis was performed using a Vybrant apoptosis assay kit to distinguish and sort out the cells undergoing early and late apoptosis and necrosis. The qualitative data shown in Figure 10 indicated that the encapsulated InsP_6_ treated MCF-7 cells had a higher population of early apoptotic cells as compared with the control cells. Etoposide used as a positive control also induced significantly higher proportion of apoptotic cell death. However, equivalent concentrations of free InsP_6_ or CS alone did not induce apoptotic cell death as compared with the control cells. These results confirm that the CS:InsP_6_ complex brings about apoptotic cell death, which was corroborated well by our above-mentioned findings on cell death observed in cytotoxicity assays and AO/EB fluorescence microscopy.

## 4. Discussion

We discovered that the encapsulation of InsP_6_ by CS, a biodegradable natural polymer, facilitates its cellular entry across semipermeable membranes and triggers apoptosis in human breast cancer MCF-7 cells. InsP_6_ is otherwise impermeable due to the presence of negatively charged phosphate groups. This study provides evidence of the incorporation of InsP_6_ into CS and its delivery into the cells. Further, we investigated the mechanism of apoptosis induced by the delivered InsP_6_, which is the production of reactive oxygen species, perhaps via intrinsic apoptotic pathways involving the mitochondria [32,54].

A vast majority of the literature suggests that the exogenously administered and endogenously manipulated InsP_6_ act as an anticancer agent that induces apoptosis in mammalian cells [32,57,58]. InsP_6_ is relatively abundant in plants, where it is primarily stored in food grains and nuts. In plants, InsP_6_ is referred to as phytic acid and is well known for its antioxidant properties. Thus, it is widely sold on the open market as a dietary supplement for human consumption and commercialized for the prevention and treatment of cancer [14]. Exogenously administered InsP_6_ has been studied in several mammalian cells, both in vitro and in vivo in animal models of colon cancer, where it displayed its potent effects as an apoptosis-inducing agent by reducing cell proliferation, cell differentiation, and tumor formation [58]. Cellular levels of InsP_6_ in most mammalian cells are present in much lower quantities than in plant cells. In a normal resting state, InsP_6_ concentrations in most mammalian cells are in micromolar (10–100 µM) concentrations [59,60,61,62]. Now, the question arises—if such levels of InsP_6_ are available in mammalian cells, how will delivering minute quantities such as those shown in this study (4 µM) have any significant impact on triggering apoptosis? Although mammalian cells synthesize their own InsP_6_ and do not ordinarily require supplemental InsP_6_, cellular InsP_6_ is available only in sufficient quantities to maintain normal cellular signaling homeostasis, including the maintenance of the balance between cell survival (proliferation) and cell death (apoptosis) pathways [58,59,62]. It is likely that mammalian cells normally do not have any extra free InsP_6_ available to rescue them under imbalanced homeostasis circumstances such as those exhibited by cancerous cells. InsP_6_ is known to be bound to InsP-binding proteins, for example, PH domain-containing proteins or divalent cations, or to be converted to other higher forms, e.g., InsP_7_ and InsP_8_ [60,61]. These scenarios are likely to reduce the chances of free InsP_6_ being available to trigger apoptosis even further. Additionally, there exists a tight and complex InsP metabolic network, which makes it even more unlikely to free up any InsP_6_ during the normal physiological state of the cell [60]. Therefore, any mechanism to deliver exogenous InsP_6_, even in small quantities, might significantly impact and alter the state of abnormal cells. To this end, researchers have been trying to find a way to deliver exogenous InsP_6_, especially to cancerous cells to trigger apoptosis, but they are concerned about its low and insufficient cellular entry from exogenous sources because of the presence of highly negative charge on InsP_6_ molecule. This study, nevertheless, represents the first of its kind to address this issue by encapsulating InsP_6_ by a biodegradable polymer to shield off its negative charge to facilitate cellular entry and trigger apoptosis.

CS, being a naturally occurring biodegradable polymer, was selected as a carrier because of its abundance in nature and established biological applications, including its role as a drug delivery vehicle [63,64,65]. This study established an ionic gelation protocol and optimized the incorporation of InsP_6_ by changing the amounts of CS to obtain various CS:InsP_6_ ratios. The CS:InsP_6_ complex was then characterized for the incorporation of InsP_6_ by two independent methods, namely PAGE analysis and AAS. The results were comparable but slightly variable with respect to the amounts of InsP_6_ detected by these two methods, perhaps due to basic procedural differences between the two methods. There is a paucity of studies with which to compare our results on the detection of InsP_6_ by these two different methods in such a scenario. The CS:InsP_6_ complex in which the InsP_6_ was incorporated the most was also characterized by physiochemical methods, viz., SEM and FTIR analyses, to establish the successful incorporation of InsP_6_ into CS. This complex was then used for further biological studies, namely of cellular uptake and apoptosis.

The cellular uptake of InsP_6_ was drastically enhanced following encapsulation with CS as compared to free InsP_6_. This study not only demonstrates that the CS:InsP_6_ complex can be successfully used as an apoptosis-inducing agent in cancerous cells, but it also provides a way to understand its mechanism of action inside the cells. It should be noted that cellular InsP_6_ has been extracted using a titanium dioxide (TiO_2_) extraction method [54] in cellular uptake studies, where TiO_2_ binds to free phosphate groups on InsP_6_ with high affinity and facilitates its extraction. Thus, it is likely that the CS:InsP_6_ complex, once it has crossed the cell membrane, releases InsP_6_, perhaps due to the biodegradation of chitosan into free InsP_6_ available to exert its biological effect. Chitosan is known to be metabolized by certain human enzymes, especially when it enters the cells [66,67]. Thus, enzymatic degradation is one of the primary mechanisms of its biodegradation. Human enzymes such as chitosanase, lysozyme, and proteases act upon the glycosidic bonds present in chitosan, hydrolyzing them into smaller fragments. Additionally, lowering the pH in endosomal and/or lysosomal compartments may also help degrade chitosan inside the cells. This establishes the utility of this nanomaterial complex to be used for exogenous administration as needed to facilitate the cellular entry of otherwise poorly permeant free InsP_6_ (Figure 6B).

As evident from cell viability, acridine orange/ethidium bromide staining, and flow cytometry analyses, the encapsulated InsP_6_ complex caused a significant increase in apoptosis at a relatively low concentration as compared to equivalent concentrations of free InsP_6_ or CS alone used to treat MCF-7 cells. The data also clearly demonstrated that cell death was mediated by early- and late-apoptotic changes and not merely due to necrosis. Additionally, this study indicated that the observed apoptotic changes were perhaps due to mitochondrial damage, as was evident by the generation of high levels of reactive oxygen species when MCF-7 cells were treated with the encapsulated InsP_6_ (Figure 9).

Previous studies, including studies from our laboratory, have investigated the mechanism of InsP_6_-mediated apoptosis in other cell types, including non-cancerous cell lines [13]. The apoptotic mechanism involved mitochondrial damage, release of cytochrome C, activation of caspase cascade, phosphorylation of AKT, a protein tyrosine kinase also known as PKB (Protein Kinase B), and DNA fragmentation. It is interesting to note that the exact abbreviation for AKT is not clear in the literature. However, it seems like AK is derived from a mouse strain and T from thymoma, a cellular source of retrovirus [68]. In this study, we limited our investigation only to human breast cancer MCF-7 cells to show its utility as an anticancer agent. Other cell models, including normal cells, can easily be tested in further studies to compare the effects of the encapsulated CS:InsP_6_ complex on MCF-7 cells. Further, InsP_6_ is known to inhibit cell proliferation and induce apoptosis by binding to the pleckstrin homology (PH) domain of AKT in the PI3K/AKT/mTOR signaling pathway [16]. The anticancer action of the CS:InsP_6_ complex might also follow a similar mechanism. It is also possible that InsP_6_, once released from the complex, exhibits other broad-spectrum activities, including cell cycle arrest and mineral-binding ability with Zn^2+^, which could affect thymidine kinase activity, essential for DNA synthesis [18,66]. It should also be noted that our in vitro experiments utilized the maximum concentration of the CS:InsP_6_ complex in a microwell plate format due to volume limitation. The use of higher concentrations of the complex might be more effective in inducing apoptosis and thus developing therapeutic applications using other experimental designs, including in vivo studies. Nevertheless, the lower concentrations of encapsulated InsP_6_ used in this study were sufficient to demonstrate its utility as an anticancer agent.

## 5. Conclusions

This study established a cellular delivery system for InsP_6_, a naturally occurring anticancer agent with poor permeability across cell membranes, by encapsulating it with chitosan, a biodegradable natural polymer. The encapsulated complex was characterized for the incorporation of InsP_6_ into CS and its cellular internalization. The complex successfully entered the MCF-7 cells and delivered InsP_6_, which induced apoptosis via ROS production at lower concentrations of InsP_6_ than previously reported using a free form of InsP_6_. This study represents the first of its kind and provides foundational work for future studies towards the goal of developing InsP-based cancer therapeutic applications.

## Figures and Tables

**Figure 1 bioengineering-11-00931-f001:**
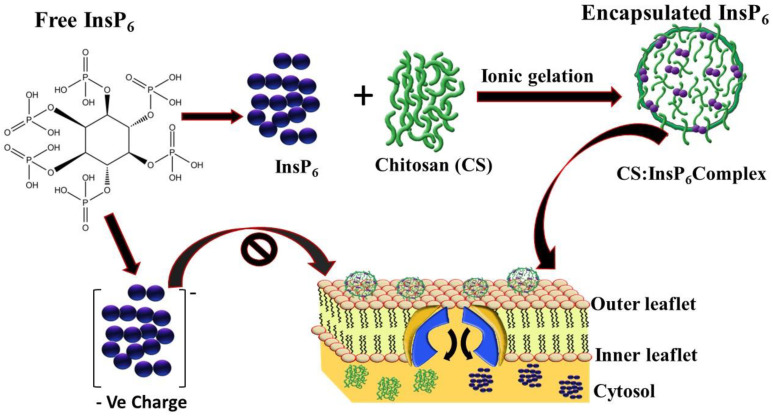
Schematic depiction of the concept of cellular entry of exogenously administered InsP_6_ after encapsulation with chitosan by ionic gelation to shield off the negative charge. Note that the encapsulated InsP_6_ enters the cell through cell membrane, whereas negatively charged free InsP_6_ is unable to enter the cell membrane.

**Figure 2 bioengineering-11-00931-f002:**
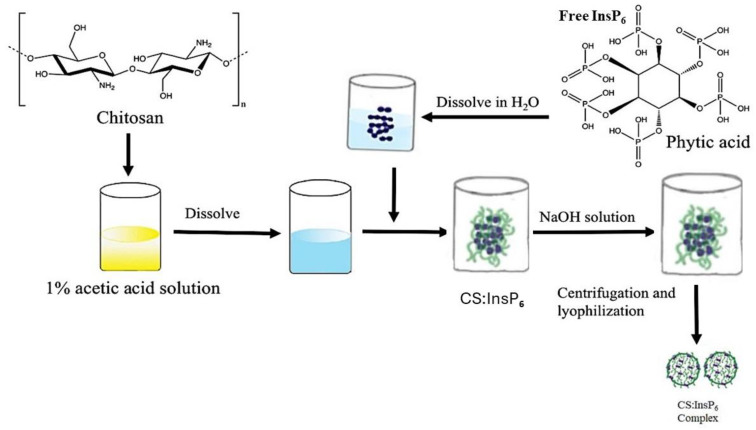
Schematic illustration of the preparation of the CS:InsP_6_ nanomaterial complex by ionic gelation. Chitosan dissolved in acetic acid (5.0 mg/mL) and InsP_6_ dissolved in deionized water (5.0 mg/mL) were mixed in varying proportions and stirred for 30 min followed by pH adjustment. The CS:InsP_6_ complex was purified by centrifugation and washing with ethanol and lyophilized to dry powder.

**Figure 3 bioengineering-11-00931-f003:**
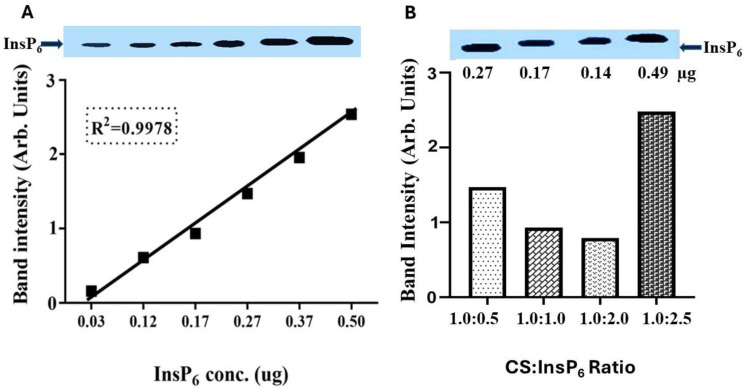
Detection of InsP_6_ contents in the CS:InsP_6_ complex by PAGE. Optimization of InsP_6_ incorporation in chitosan was carried out by varying the ratios of CS:InsP_6_ (**B**). Standard InsP_6_ with known concentrations were also run in parallel to establish the linearity of detection (**A**). Band densities were analyzed by image J software. (**B**) shows the amounts of InsP_6_ detected in the samples with various ratios of CS:InsP_6_ applied on the gel. The maximum amount of InsP_6_ (0.49 μg) was detected in the sample with a CS:InsP_6_ ratio of 2.5:1.0. This amount (0.49 μg), when calculated using the dilution factor of the samples loaded on the gel, provides a total incorporation of 49 ug InsP_6_ per mg CS. Data shown are a representative of at least three independent experiments with similar results. The error bars are not shown as the data shown are from a single experiment repeated at least three times.

**Figure 4 bioengineering-11-00931-f004:**
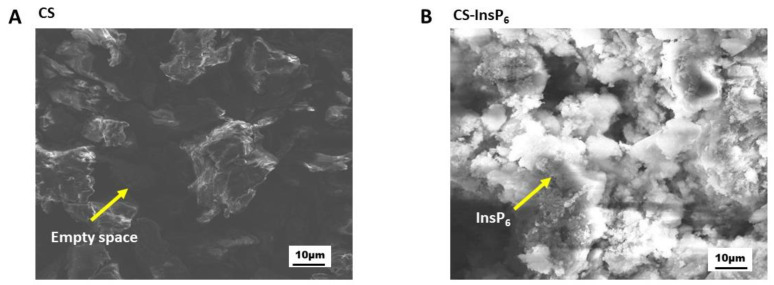
SEM images of CS (**A**) and the CS:InsP_6_ complex (**B**). Arrows point to the empty spaces in CS (**A**) that were perhaps filled by InsP_6_ (**B**), showing morphological changes following encapsulation. The CS:InsP_6_ complex at a ratio of 2.5:1.0 was used for SEM analysis in (**B**). Electron micrographs shown are representative images seen in replicate experiments with similar results.

**Figure 5 bioengineering-11-00931-f005:**
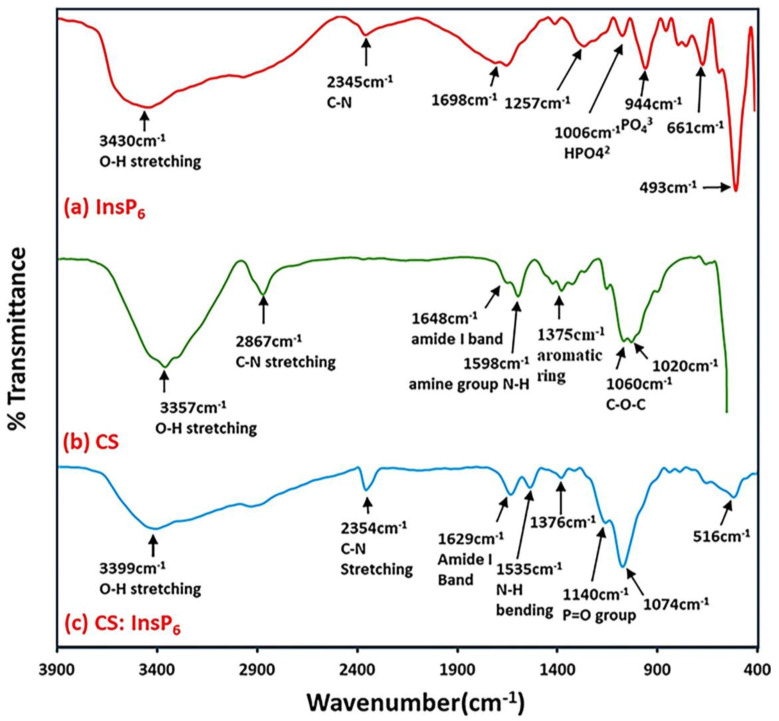
FTIR spectra of (**a**) InsP_6_, (**b**) CS, and (**c**) encapsulated complex with a CS:InsP_6_ ratio of 2.5:1.0. Note that the spectral properties of the characteristic bands at specific wavenumbers in InsP_6_ (**a**) and CS (**b**) are changed upon encapsulation (**c**).

**Figure 6 bioengineering-11-00931-f006:**
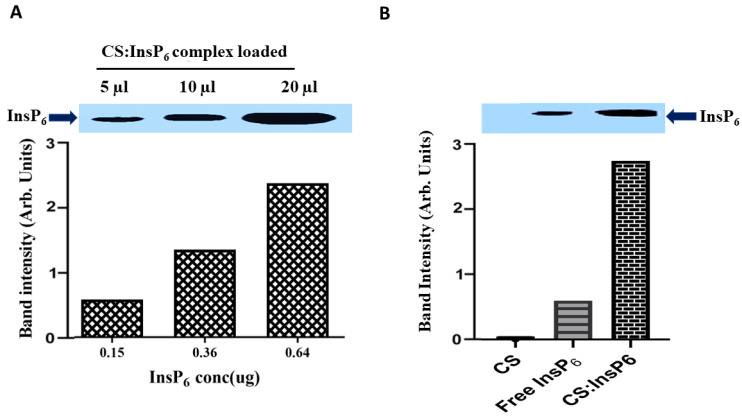
Cellular uptake of encapsulated InsP_6_. Band intensity was analyzed by image J software. (**A**) shows quantitative detection of InsP_6_ in the CS:IsP_6_ complex with a CS:InsP_6_ ratio of 2.5:1.0. A volume of 5, 10, and 20 µL of the complex loaded on the gel gave 0.15, 0.36, and 0.64 µg InsP_6_, respectively, showing a concentration-dependent linear increase in the detection of InsP_6_ in the complex. (**B**) shows a significant increase in InsP_6_ uptake by MCF-7 cells using the encapsulated complex with a CS:InsP_6_ ratio of 2.5:1.0 as compared to corresponding free InsP_6_ and CS. Data shown are representative of experiments performed independently at least three times with similar results. Statistical analysis is not shown as the data are from a representative experiment.

**Figure 7 bioengineering-11-00931-f007:**
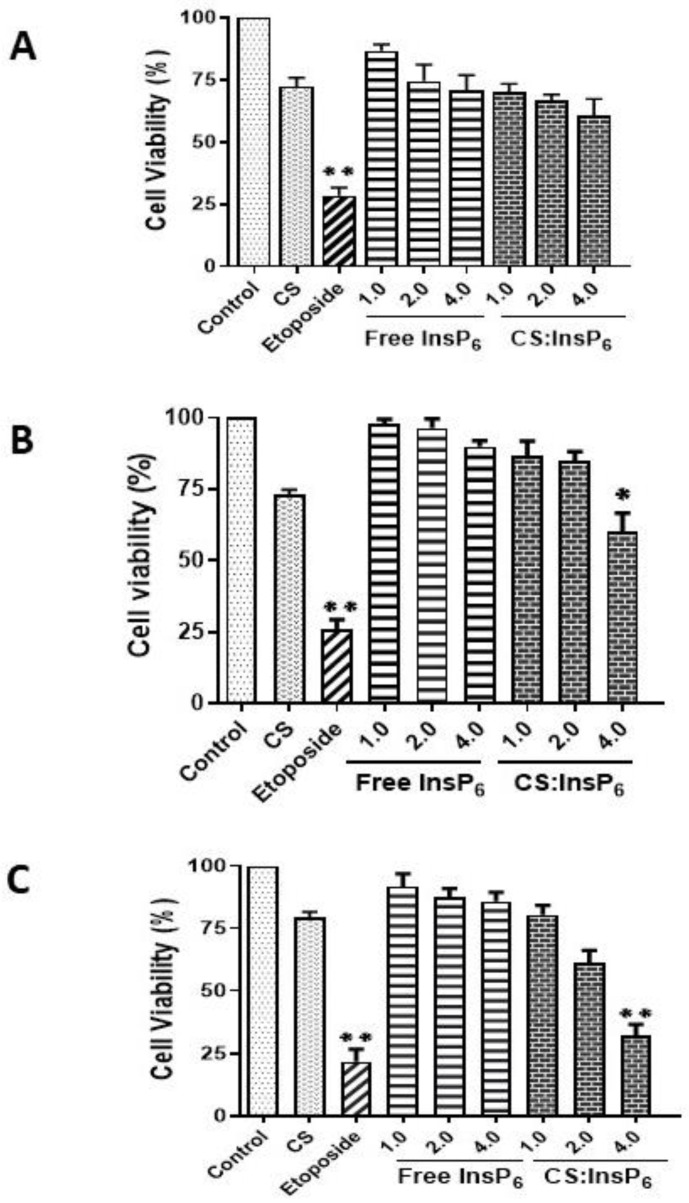
Dose- and time-dependent induction of cell viability in MCF-7 cells by encapsulated InsP_6_ treatment. Cell viability was determined at 24 h (**A**), 48 h (**B**), and 72 h (**C**) by MTT assay using the given doses of free InsP_6_ (1.0–4.0 µM) and equivalent amounts of the CS: InsP_6_ complex that would give similar doses of free InsP_6_. Data are shown as means ± SD from three independent experiments. All experimental values were statistically compared with their respective controls to determine any significant differences. Only treatment with 4 µM encapsulated InsP_6_ gave a significant difference as compared with 4 µM free InsP_6_. * *p* value ≤ 0.001 or ** *p* ≤ 0.0001 show significantly different values as compared to the respective controls.

**Figure 8 bioengineering-11-00931-f008:**
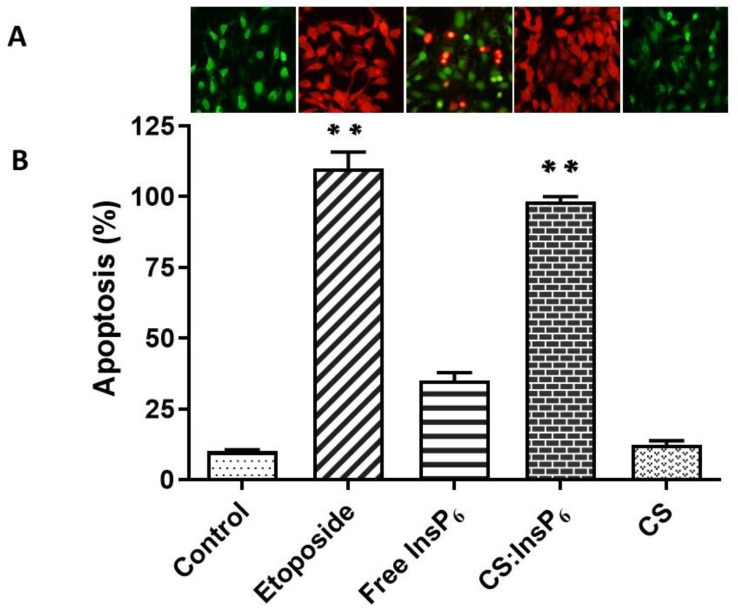
Effect of encapsulated InsP_6_ on apoptosis. MCF-7 cells were incubated with 4 µM of encapsulated InsP_6_ for 72 h to induce apoptosis. Etoposide (100 µM) was used as a positive control. (**A**) MCF-7 cells were stained with acridine orange/ethidium bromide and visualized under UV light using a fluorescent microscope. (**B**) The percentage of apoptosis was determined by counting 200–300 live (green) and/or dead (red) cells. Values shown are mean ±SD from three experiments, each performed in triplicate. ** *p* value ≤ 0.001 as compared to the control.

**Figure 9 bioengineering-11-00931-f009:**
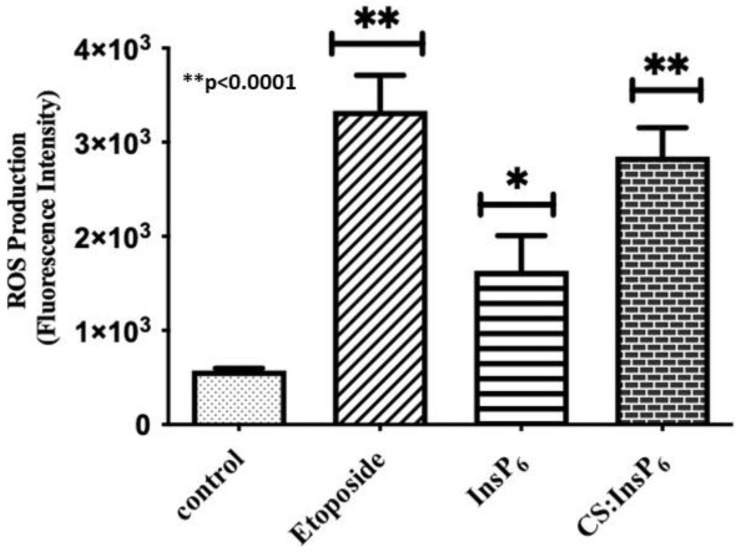
Effect of encapsulated InsP_6_ on ROS generation. MCF-7 cells were treated with 4 µM free InsP_6_ equivalent of the encapsulated CS:InsP_6_ complex for 72 h in a 96-well microplate. Etoposide (100 µM) was used as a positive control. Cells were then stained with 10 µM DCFH-DA and fluorescence intensity was recorded using a fluorescence microplate reader. Values shown are mean ± SD from three independent experiments, each performed in triplicate. * *p* value of ≤0.0001 was considered significantly different compared to the control.

**Figure 10 bioengineering-11-00931-f010:**
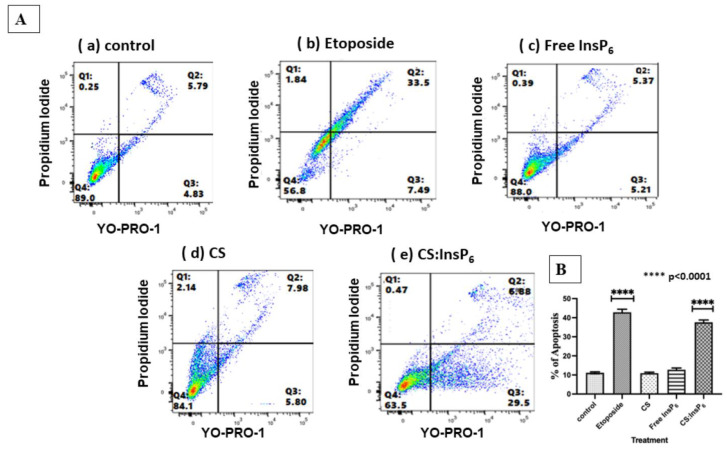
Determination of specificity of encapsulated InsP_6_-induced apoptosis by flow cytometry (**A**). Apoptosis was measured by using a commercially available Vybrant apoptosis assay kit #4. Live cells - are shown as green in lower left quadrant and apoptotic cells are shown as blue in **lower and upper right** quadrant. Necrotic cells give a red color which are expected to show up in upper left quadrant. The data shown are representative of an experiment repeated at least three times with similar results. (**B**) shows statistical analysis results of the flow cytometry data showing mean ± standard deviation (SD) from three independent experiments. The % apoptosis values were obtained by combining early and late apoptosis values from the lower and upper right quadrants, respectively. One-way ANOVA with multiple comparisons was used to determine values that were statistically significant. **** *p* < 0.0001 was considered statistically significant values compared with their respective controls.

## Data Availability

Data are contained within the article.

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
