# Peer review of "Encapsulation of Inositol Hexakisphosphate with Chitosan via Gelation to Facilitate Cellular Delivery and Programmed Cell Death in Human Breast Cancer Cells"

_bioengineering, 2024, doi:10.3390/bioengineering11090931_

Round 1
Reviewer 1 Report
Comments and Suggestions for Authors
In this manuscript, the authors have demonstrated the effective delivery of drugs into cells by encapsulating InsP6 with Chitosan, resulting in the generation of reactive oxygen species (ROS) within the cells and inducing apoptosis. The authors have provided rigorous experimental evidence to support these findings.
However, there are several issues that need to be addressed:
-
Figure 3B: There is an error in the description of the results. The figure shows the CS:InsP6
ratio as 1.0:2.5, while the figure caption states 2.5:1.0. Additionally, the y-axis label in Figure 3B should be added. -
Figure 4: The SEM results do not clearly show the binding between InsP6 and Chitosan. Please include TEM elemental mapping images to verify the presence of phosphorus (P) in the CS-InsP6 complex, confirming the binding of InsP6 with Chitosan.
-
Figure 9: Please present the DCFH-DA fluorescence images for the different groups.
-
Figure 10: Please provide the apoptotic statistics for the flow cytometry results.
Author Response
Point by Point Responses to Reviewer comments
Reviewer 1
- Figure 3B: There is an error in the description of the results. The figure shows the CS:InsP6 ratio as 1.0:2.5, while the figure caption states 2.5:1.0. Additionally, the y-axis label in Figure 3B should be added.
Response: Authors thank the reviewer for pointing out. We have revised Figure 3B with correct CS:InsP6 ratios and added the Y-axis, as suggested by the reviewer. Figure captions are also updated.
- Figure 4: The SEM results do not clearly show the binding between InsP6 and Chitosan. Please include TEM elemental mapping images to verify the presence of phosphorus (P) in the CS-InsP6 complex, confirming the binding of InsP6 with Chitosan.
Response: Authors agree with the reviewer that SEM images do not clearly show the binding between InsP6 and chitosan. Nevertheless, SEM images show some morphological changes following encapsulation. Authors express inability to perform TEM elemental mapping images, as this facility is not available to the authors at this time. However, we have modified the description in the results section on page 9 to explain that the result on SEM analysis led us to analyze the complex by FTIR spectral analysis which confirmed the presence of phosphorous in the CS:InsP6 complex (Figure 5).
- Figure 9: Please present the DCFH-DA fluorescence images for the different groups.
Response: Figure 9 is based on the quantitative measurement of the intensity of DCFH-DA fluorescence by a microplate reader. The way this assay is performed does not provide any fluorescence images of the cells and hence the fluorescence images are not given for the different groups. However, we have revised Figure 9 to appropriately label the Y-axis and determined statistical differences with respect to different groups.
- Figure 10: Please provide the apoptotic statistics for the flow cytometry results.
Response: Authors agree with reviewer. As suggested, a new figure 10B with its figure legends is added based on statistical results from three flow cytometry experiments.
Reviewer 2 Report
Comments and Suggestions for Authors
In this work, the authors report the cellular delivery of InsP6 in cancerous cells wherein InsP6 was encapsulated by chitosan (CS), a natural polysaccharide, by ionic gelation method. The hypothesis was that encapsulated InsP6 will enter the cell more efficiently to trigger its apoptotic effects. The incorporation of InsP6 into CS was optimized by varying the ratios of the two and confirmed by InsP6 analysis by polyacrylamide gel electrophoresis (PAGE) and Atomic Absorption Spectrophotometry (AAS). The complex was further characterized by Scanning Electron Microscopy (SEM) and Fourier Transform Infrared Spectroscopy (FTIR) for physicochemical changes. The data indicated morphological changes as well as changes in spectral properties of the complex upon encapsulation. The encapsulated InsP6 enters human breast cancer MCF-7 cells more efficiently than free InsP6 and triggers apoptosis via a mechanism involving the production of reactive oxygen species (ROS). This work has potential for developing cancer therapeutic applications utilizing natural compounds that are likely to overcome severe toxic effects associated with synthetic chemotherapeutic drugs.
While it was good attempt to address cellular delivery of InsP6 in cancerous cells using chitosan, I would like to seek major clarification on the following points before considering the work for publication in the journal.
1) In the discussion section, the authors have noted that ‘Cellular levels of InsP6 in most mammalian cells range in micromolar (10-100 µM) concentrations. However, inducing apoptosis with such lower concentrations in these cells is often challenging to maintain cellular apoptotic homeostasis.’ In this work, the role of chitosan is to act as a delivery medium to facilitate the transfer of InsP6 into the cell. However, if the authors state that previous reports have failed to induce cell death with (10-100 µM) concentrations of InsP6 then in such a scenario how is 4 µM encapsulated and delivered through chitosan in the current work able to induce cell death?
2) In Figure 3B, the authors have cited 0.49 µg of InsP6 to be the highest concentration loaded however, in the text, 49 µg InsP6 per mg chitosan was cited as the highest. Please cross-check the data and clarify.
3) In Figure 5, the authors are advised to put a co-plot/combined FTIR so that it is easier to compare the peak intensity and peak shifts.
4) In Figure 7, please label A,B,C in the figure captions. Also, the authors should provide the statistical significance of encapsulated InsP6 with respect to free InsP6 in different concentrations and not just compare InsP6 with the control.
5) The authors have also stated that ‘it is likely that the CS:InsP6 complex once crosses the cell membrane releases InsP6, perhaps due to biodegradation of the chitosan, into free InsP6 available to exert its biological effect.’ What are the likely conditions that might lead to the biodegradation of chitosan?
6) In the ‘Cellular Uptake of Encapsulated InsP6 by MCF-7 Cells’ section why did the authors use CS:InsP6 ratio: 2.5:1.0? There seems to be no supporting loading data corresponding to this ratio?
7) It is highly advisable that the authors should report all data in the same units so as to give a uniform review of the results. The loading and characterization of InsP6 into CS were reported in µg whereas abruptly the notations were changed to µM from the section ‘Effect of encapsulated InsP6 on apoptosis in MCF-7 cells’ onwards. This makes it difficult to make a fair comparison of the results obtained.
Author Response
Point by Point Responses to Reviewer comments
Reviewer 2
1) In the discussion section, the authors have noted that ‘Cellular levels of InsP6 in most mammalian cells range in micromolar (10-100 µM) concentrations. However, inducing apoptosis with such lower concentrations in these cells is often challenging to maintain cellular apoptotic homeostasis.’ In this work, the role of chitosan is to act as a delivery medium to facilitate the transfer of InsP6 into the cell. However, if the authors state that previous reports have failed to induce cell death with (10-100 µM) concentrations of InsP6 then in such a scenario how is 4 µM encapsulated and delivered through chitosan in the current work able to induce cell death?
Response: Authors thank the reviewer for raising such a concern which is related to the regulation of normal cellular homeostasis by InsP6. We looked at the discussion more closely and realized that perhaps we initially failed to explain this aspect in enough detail. We have now modified the discussion and explained the relevance of cellular delivery of exogenous InsP6 even in the presence of lower concentrations of endogenous InsP6.
Briefly, although mammalian cells synthesize their own InsP6 and do not ordinarily require supplemental InsP6, the cellular InsP6 is available only in sufficient quantities to maintain normal cellular signaling homeostasis including the maintenance of the balance between cell survival (proliferation) and cell death (apoptosis) pathways. Cellular levels of InsP6 present in mammalian cells are in much lower quantities than in plants cells and they are representative of the normal resting state of the cells. Additionally, cellular InsP6 is known to exist in bound forms such as with PH domain containing proteins, divalent cations or converted to other higher InsPs e.g., InsP7 and InsP8 which make it less likely for any free InsP6 available to additional biological functions. Further, the literature suggests that there exists a tight and complex InsPs metabolic network which makes it even unlikely to free up any InsP6 during the normal physiological state of the cell. These scenarios are likely to reduce any chances of free InsP6 available to trigger apoptosis to rescue cells under imbalanced homeostasis circumstances such as those exhibited by cancerous cells. Hence, any extra InsP6 delivered to the cancerous cells might help trigger more apoptosis.
2) In Figure 3B, the authors have cited 0.49 µg of InsP6 to be the highest concentration loaded however, in the text, 49 µg InsP6 per mg chitosan was cited as the highest. Please cross-check the data and clarify.
Response: 0.49 µg InsP6 shown in Figure 3B is the absolute amount of InsP6 detected on the gel based on the band density of the sample in the volume loaded on the gel. This value was then converted to µg InsP6 per mg of CS using the dilution factor of the sample loaded on the gel which turns out to be 49 µg InsP6 per mg CS. This has now been explained in detail in the results section on page 8 and in the figure legends for Figure 3B.
3) In Figure 5, the authors are advised to put a co-plot/combined FTIR so that it is easier to compare the peak intensity and peak shifts.
Response: Authors thank the review for this suggestion and agree with the comment. As suggested, we have combined the FTIRs and a new plot of the wavenumber Vs. % transmittance is provided in Figure 5.
4) In Figure 7, please label A,B,C in the figure captions. Also, the authors should provide the statistical significance of encapsulated InsP6 with respect to free InsP6 in different concentrations and not just compare InsP6 with the control.
Response: Authors thank the reviewer for this suggestion. We have labelled the figure captions as suggested and modified the text in results on page 12 and in figure captions of Figure 7 to reflect any statistical differences by comparing experimental values with their respective controls at all concentrations of InsP6.
5) The authors have also stated that ‘it is likely that the CS:InsP6 complex once crosses the cell membrane releases InsP6, perhaps due to biodegradation of the chitosan, into free InsP6 available to exert its biological effect.’ What are the likely conditions that might lead to the biodegradation of chitosan?
Response: Authors thank the review for raising this interesting issue on biodegradation of chitosan once it enters human cells. Existing literature indicates that enzymatic degradation is one of the primary mechanisms responsible for its biodegradation inside human cells. Human enzymes such as chitosanase, lysozyme, and proteases act upon glycosidic bonds present in chitosan, hydrolyzing them into smaller fragments. Additionally, lowering of the pH in endosomal/ lysosomal compartments also helps degrade chitosan inside the cells. We have explained this briefly in the discussion on page 17 and provided a couple of references.
6) In the ‘Cellular Uptake of Encapsulated InsP6 by MCF-7 Cells’ section why did the authors use CS:InsP6 ratio: 2.5:1.0? There seems to be no supporting loading data corresponding to this ratio?
Response: Authors thank the reviewer for pointing this out. However, we believe that this comment arose as a result of our miss mislabeling of Figure 3B in the original manuscript where we analyzed the incorporation of InsP6 into CS by loading the varying ratios of the CS:InsP6 on to the gel. In fact, we analyzed the CS:InsP6 ratio of 2.5:1.0 and this ratio gave the maximum incorporation of InsP6. This is why we used this ratio not only for cellular uptake studies but also for physical characterization by SEM and FTIR, as well as in apoptosis studies. Figure 3B is now revised and the loading ratios of the CS:InsP6 complexes are accurately labelled.
7) It is highly advisable that the authors should report all data in the same units so as to give a uniform review of the results. The loading and characterization of InsP6 into CS were reported in µg whereas abruptly the notations were changed to µM from the section ‘Effect of encapsulated InsP6 on apoptosis in MCF-7 cells’ onwards. This makes it difficult to make a fair comparison of the results obtained.
Response: Authors thank the reviewer for the advice to maintain the uniformity of the units and we appreciate it. However, the µg quantities are only expressed for the analysis of InsP6 by gel electrophoresis to compare the band densities and absolute amounts with the known standard InsP6 used in terms of µg and run simultaneously. At that time, it was easier to compare using the µg units in gel electrophoresis. We then calculated molar concentration to study biological effects in all cell-based assays including MTT, ROS production and apoptosis experiments. It would require enormous effort to redo some of the work to bring all values to the same units at this point. Additionally, it will not change any results and conclusion. However, we have ensured in the manuscript that it does not cause any confusion for the readers.
Reviewer 3 Report
Comments and Suggestions for Authors
The manuscript requires some adjustments to be published, briefly presented in the attached document

Author Response
Point by Point Responses to Reviewer comments
Reviewer 3
- There are some abbreviations in the text that are not explained: AKT, TBE, APS, TEMED, HBSS, ….
Response: All abbreviations are thoroughly checked and explained where they first encountered. However, it is interesting to note that the exact abbreviation for AKT, a protein tyrosine kinase, also known as PKB (Protein Kinase B) is not clear in the literature. It seems like AK is derived from a mouse strain and T from thymoma, a cellular source of retrovirus. We have modified the text in “Discussion” section to explain this and provided a new reference to this source of information.
- There is a certain discrepancy in the bibliographic references, I don't know exactly where it started, the authors must carefully check the references. For example, lines 87-89 "However, attempts were made to deliver negatively charged inorganic polyphosphates (PolyP) using amphipathic guanidinium-rich polycarbonates as nanocarriers with an emphasis to understand their biological relevance [37]" - in this case the reference [37] is actually [36]. Likewise below, reference [38] should be replaced by [37] and [39] by [38]. The references must be checked, so that the citation is correct.
Response: Authors agree with the reviewer’s comment. There were some mistakes made in the reference numbers cited in the text and corresponding number listed in References. This has altered the appropriateness of the references. Now we have checked all references thoroughly and corrected them where needed. We have also added some new references and replaced others.
- Line 143 “acidic acid”?
Response: Typographical error corrected. Thank you for pointing this out.
- Line 147: "volumetric ratios of CS:InsP6 of 0.5:1.0, 1.0:1.0, 2.0:1.0, and 2.5:1.0 (v/v)" - I don't know if the ratios are written correctly. The authors kept constant the volume of InsP6 (1.0) and increased the volume of CS from 0.5 to 2.5? Or, in fact, the volume of CS is constant (0.1) and the volume of InsP6 increases from 0.5 to 2.5? If the last version is correct, then the ratios should be reversed to be correct, as it is very well mentioned in Fig. 3B.
Response: Authors agreed with the reviewer’s comment that there was a mistake made in labelling of the volumetric ratios of CS:InsP6 in Figure 3B. In fact, the volumetric ratios of CS:InsP6 are 0.5:1.0, 1.0:1.0, 2.0:1.0, and 2.5:1.0 (v/v) as described in the text and in the legend of Figure 3B. However, there was a mistake made in labeling the ratios in the x-axis of Figure 3B. A new Figure 3B is now added with correct ratio. Additionally, we have also labelled Y-axis in Figure 3B.
- Line 235: “Cells were then treated with various concentrations of CS:InsP6 complex and CS” – wasn’t InsP6 free also used?
Response: Authors agree with the reviewer’s comment. Yes, various concentrations of free InsP6 were also used as a control. The sentence has now been modified to reflect the use of free InsP6 on page 6 in methods section, 2.8. MTT Assay.
- Lines 283-284: must be explained what "YO-PRO-I stock solution" and "PI stock solution" consist of?
Response: YO-PRO-1 stock solution (component A) consists of 100 µM YO-PRO-1 in DMSO. It is a green fluorescence dye that can enter apoptotic cell membranes. Whereas Propidium Iodide (PI) stock solution (component B) consists of 1 mg PI /mL (1.5 mM) solution in deionized water. These stock solutions are explained on page 7, methods section 2.11: Flow Cytometry.
- Figure 3: the Y axis must also be added to Fig. 3B, even if it is understood that it is the same as in Fig3A. What does "0.49 ug" mean? "CS:IsP6 ratio of 2.5:1.0" is actually "CS:InsP6 ratio of 1.0:2.5" to be correct.
Response: A new Figure 3B is added with Y-axis labelled. Also, the ratios of CS:InsP6 were labeled wrong, but these are now fixed. The ratios of CS:InsP6 on X-axis are 0.5:1.0, 1.0:1.0, 2.0:1.0 and 2.5:1.0. These ratios represent the samples loaded on the gel that gave 0.27, 0.17, 0.14, and 0.49 mg InsP6, respectively, based on the band density as calculated using standard InsP6. So, 0.49 mg is the amount of InsP6 calculated for the CS:InsP6 complex with the ratio of 2.5: 1.0. So, this value was used to calculate the maximum incorporation of InsP6 per mg of chitosan which turned out to be 49 ug per mg of chitosan factoring the dilution factor of the sample loaded on the gel.
- Line 338: “CS:InsP6 complex appeared filled” - it should be mentioned which of the four complexes obtained was analyzed by SEM, what is the CS:InsP6 ratio? Likewise in the case of FTIR analysis (line 348)
Response: Authors agree with the reviewer’s viewpoint on the missing information on the ratio of CS:InsP6 in the complex used for the analysis by SEM and FTIR. The ratio of the CS:InsP6 in the encapsulated complex used for the analysis by SEM, FTIR as well as for cellular uptake was 2.5:1.0. This information has now been updated in Figure legends for Figure 5, 6 and 7 and also in the text on page 8 in Results section 3.1.
- Figures 5: the figures should be enlarged, you can't see the peaks well; Similarly, Figures 7 are too small compared to other figures in the manuscript.
Response: Authors appreciate the suggestion and agree with the reviewer’s comment. Figure 5 is redrawn with plots for CS, InsP6 and CS:InsP6 co-plotted and enlarged. Figure 7 is also enlarged.
- The authors cite references older than the year 2000! Are those references necessary? Can't they be replaced with newer ones?
Response: Authors thank the reviewer for suggestion to updating any references older than the year 2000. We have replaced references older than the year 2000 with newer references.
Round 2
Reviewer 2 Report
Comments and Suggestions for Authors
All issues have been addressed. Can be considered for publication in the present form.